# Coevolution of Hydrological Cycle Components under Climate Change: The Case of the Garonne River in France

**Youen Grusson [1,2,3,\*]** , **François Anctil [2,\*]**, **Sabine Sauvage [3]** and **José Miguel Sánchez Pérez [3]**

1   Department of Soil and Environment, Swedish University of Agricultural Sciences
    (SLU—Sveriges lantbruksuniversitet), PO Box 7014, SE-750 07 Uppsala, Sweden
2   Department of Civil and Water Engineering, Université Laval, Québec, QC G1V 0A6, Canada
3   EcoLab, University of Toulouse, CNRS, INPT, UPS, Avenue de l'Agrobiopole, 31326 Castanet-Tolosan,
    France; sabine.sauvage@univ-tlse3.fr (S.S.); jose-miguel.sanchez-perez@univ-tlse3.fr (J.M.S.P.)
*   Correspondence: youen.grusson@slu.se (Y.G.); Francois.Anctil@gci.ulaval.ca (F.A.)

**Abstract:** Climate change is suspected to impact water circulation within the hydrological cycle at catchment scale. A SWAT model approach to assess the evolution of the many hydrological components of the Garonne catchment (Southern France) is deployed in this study. Performance over the calibration period (2000–2010) are satisfactory, with Nash–Sutcliffe ranging from 0.55 to 0.94 or $R^2$ from 0.86 to 0.98. Similar performance values are obtained in validation (1962–2000). Water cycle is first analyzed based on past observed climatic data (1962–2010) to understand its variations and geographical spread. Comparison is then conducted against the different trends obtained from a climate ensemble over 2010–2050. Results show a strong impact on green water, such as a reduction of the soil water content (SWC) and a substantial increase in evapotranspiration (ET) in winter. In summer, however, some part of the watershed faces lower ET fluxes because of a lack of SWC to answer the evapotranspiratory demand, highlighting possible future deficits of green water stocks. Blue water fluxes are found significantly decreasing during summer, when in winter, discharge in the higher part of the watershed is found increasing because of a lower snow stock associated to an increase of liquid precipitation, benefiting surface runoff.

**Keywords:** water cycle; climate change; SWAT modelling; coevolution of water resources; blue and green water

---

## 1. Introduction

Climate change has become an important factor in water management and planning [1]. Its impact on water resources must therefore be studied along regional and national policies. Numerous studies already assessed this impact on various hydrological components of the hydrologic cycle. In Europe, for instance, several reports [2,3] attest lesser water volumes in rivers; a conclusion notably shared by the IPCC AR5 report [1,4]. The documented trend was found unequally distributed over the year: volumes are increasing in winter but not to the point of compensating for a general diminution the rest of the year [5]. The general hydrological regime of rivers is thus evolving [6,7], including a change in frequency and magnitude of extreme events. Still, discharge is not the only component of the water cycle to be impacted by climate change. The cryosphere, which locally dictates streamflow [8], is declining in volume and extent across the European continent. For example, a lost from 4 to 10 cm of snowfall per decade have been identified over the last century [9–11]. Groundwaters are impacted as well but, in this instance, it is particularly difficult to separate the role of climate

change from other anthropogenic interferences [12–14]. Their recharge is nonetheless expected to decrease over Europe, with the exception of a short winter period in the northern parts of the continent [15]. Evapotranspiration processes are also altered [16–20], leading to changes in the soil water content [21,22]. All these processes being intertwined, it appears obvious that any modification on one component could be transmitted to the others. For instance, the evolution of the soil water content is strongly linked to changes in evapotranspiration [20] and groundwater recharge that in turn modulates discharge intensities [23].

Most of the above studies focused on a single component of the hydrological cycle. Falkenmark, et al. [24,25] pointed out the weaknesses of such a siloed approach and recommended encompassing all hydrologic components at once. They suggest moving from a focus on surface water and aquifer (blue water) to a broader focus that would encompass also evapotranspiration or soil moisture (green water) [26]. This consideration is yet unaddressed in most hydrological studies. For instance, in a recent article about climate water scarcity, Mekonnen, et al. [27] held forth on the limit of strictly considering blue water. A spatio-temporal assessment exploring the co-evolution of all major components of the hydrological cycle is deemed preferable. Of course, some published works did examine blue and green waters (e.g., Zang, et al. [28], Zang, et al. [29], Gosain, et al. [30], Faramarzi, et al. [31], Abbaspour, et al. [32], Schuol, et al. [33], Schuol, et al. [34]) but mostly as non-interacting categories. Zuo, et al. [35] went further than most and simulated the spatiotemporal variability of green and blue waters at regional scale. They concluded on the dissimilar evolution of both types of water, identifying a need for differentiated management practices. Climate change remained however unaddressed in their analysis. In short, many advocate exploring the coevolution of all major components of the water cycle (and their different water paths) but such endeavor is still rarely conducted.

This study has for objective to propose a methodology for an integrated analysis of the coevolution of all major components of the hydrological cycle under climate change. The innovative approach developed in this paper considered past and future climates, covering a 100-year period. The use of those two datasets to feed the agro-hydrological model SWAT over one of the major watershed of France (the Garonne river watershed) has led to the simulation of those major hydrological components. This double approach has allowed to interpret projected futures impacts on water resources regarding changes that have already occurred during the last 50 years.

## 2. Materials and Methods

### 2.1. Study Site

The Garonne River is a 525-km fluvial system located in the southwest of France. Its watershed drains 55,000 km$^2$ to the Atlantic Ocean. It extends over three main geographic entities: the Pyrenees to the south where some peaks exceed 3000 m, the plateau of the Massif Central to the northeast that reaches up to 1700 m, and the plain between them which elevation is less than a few hundred meters (Figure 1). Different hydrological behaviors result from this diverse landscape and associated altitudes and slopes [36]. The present study simulates the Garonne River watershed down to Tonneins gauging station: the lowest gauging station uninfluenced by the tides (Figure 1), collecting water from some 50,000 km$^2$ of land. The Tonneins station experienced an average streamflow of 603 m$^3$/s from 1913 to 2013. The highest value on record reached 5700 m$^3$/s and the lowest, 37.5 m$^3$/s. The highest interannual monthly flow occurred in February (970 m$^3$/s) and the lowest, in August (177 m$^3$/s http://www.hydro.eaufrance.fr/).

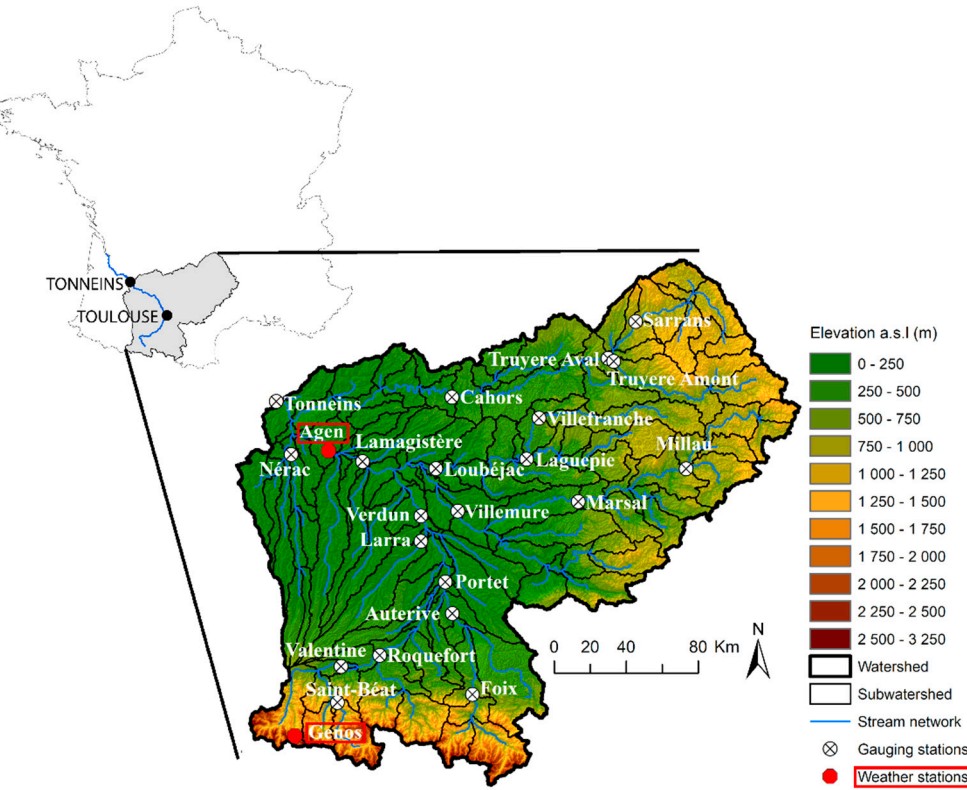

**Figure 1.** Garonne watershed.

The Garonne watershed is distinctive by its topographic and climatic diversity. In mountainous uplands, temperatures fall frequently below freezing in winter, while the plain rarely experiences below zero temperatures. An analysis of weather data provided by Météo-France (https://donneespubliques.meteofrance.fr) depicts this variability. From 2000 to 2010, at the Genos gauging station (1,250 m a.s.l.; Figure 1) reported mean monthly temperatures of −3 °C in February and of 20 °C in August. Over the same period, the Agen gauging station (58 m a.s.l.; Figure 1) recorded 2.5 °C in February and 29 °C in August. A similar disparity exists in total precipitation: 1,544 mm at Genos, but 649 mm at Agen. Land use is also quite diverse: forest and alpine grassland abound in mountains (37% of the watershed), while the plain is dominated by agricultural activities (60%) [37]. The remaining uses are either artificialized land (2.5%) or water bodies (0.5%). Cambisols are the most frequent soil classes, covering 55% of the watershed. Different classes of shallow soils are also present in the upper part of the watershed (19%), such as lithosols, regosols, andosols, rendzinas, and rankers. Luvisols cover 18% of the watershed, mostly on hillsides and in the plain. The river itself mostly flows on fluvisols (7%). The remaining 1% consists of anecdotal types: arenosols, podzol, histosol, and artificialized soil.

Human activities impact the watershed mainly by the presence of several dams in the upper portion of the river. They are mostly designed for low flow support and irrigation purposes. Sauquet, et al. [38] and Hendrickx, et al. [39] compared observed and naturalized discharge data for several sites within the watershed. They highlighted that only the St. Beat and Valentine gauging stations (Figure 1), in the upper part of the basin, are significantly impacted by human activities at a monthly time-step. This translates into lower flows in spring and higher ones in winter. Since dams are mostly located on smaller streams in the mountainous upper part of the basin, contributions from larger, less affected streams quickly diminish the anthropic influence of dams, which is barely noticeable in the valley.

## 2.2. SWAT Model Implementation

SWAT is an agro-hydrological semi-distributed model that requires an areal discretization process that consists in dividing the watershed into sub-watersheds based on the river network and topography [40–42]. The implementation procedure starts by identifying hydrological response units (HRUs) within each subwatershed: areas deemed homogeneous after compiling soil, land cover, and slope information. HRUs are the base unit to compute the water balance articulated around four compartments—snow, soil, shallow aquifer, and deep aquifer—linked by hydrological processes such as infiltration, runoff, evapotranspiration, lateral flow, and percolation. Computation is performed at the HRU level, aggregated at subwatershed level, and routed from upstream to the final outlet by streams. The project exploited in this study has been set up with ArcSWAT 2012, a GIS-based graphical interface, helping users to define HRUs and generate the associated input files [43]. In our modeling setup, management operations are kept into automatic mode: fertilizer is applied when plants experience nitrogen stress, and irrigation is conducted when water stress is detected over irrigated agricultural land.

Data sources are gathered in Table 1. ASTER product from NASA is used for topographic data, while soil and land use information are taken from the European Soil Data Base (ESDB) and Corinne Land Cover (CLC).

**Table 1.** Data sources.

| Data Type | Data Source | Scale |
|---|---|---|
| DEM | NASA/METI [44] | 90 × 90 m |
| Land cover | Corine Land Cover [37] | 1:100,000 |
| Soil | European Soil Database [45] | 1:1,000,000 |
| Climate (SAFRAN) | Météo-France https://donneespubliques.meteofrance.fr/) | 8 × 8 km |
| Climate (Projection) | DRIAS (http://www.drias-climat.meteo.fr) | 8 × 8 km |
| River discharge | Banque Hydro (http://www.hydro.eaufrance.fr/) | Daily |

Discharge data are taken from 21 gauging stations spread over the watershed (Figure 1) and covering a period from 1962 to 2010. Selection have taken into account the hydrological and morphological diversity of the watershed [36] and the length of the time series. Availability is given in Table 2 for each station.

**Table 2.** Available chronicle for selected gauging stations.

| Station Name | Date | Station Name | Date | Station Name | Date |
|---|---|---|---|---|---|
| Saint-Béat | 1971–2010 | Verdun | 1972–2010 | Lamagistère | 1967–2010 |
| Valentine | 1962–2010 | Millau | 1969–2010 | Sarrans | 1962–2010 |
| Roquefort | 1962–2010 | Marsal | 1962–2010 | Truyère Amont | 1972–2010 |
| Foix | 1962–2010 | Villemure | 1970–2010 | Truyère Aval | 1979–2010 |
| Auterive | 1966–2010 | Villefranche | 1962–2010 | Cahors | 1962–2010 |
| Portet | 1962–2010 | Laguepie | 1962–2010 | Nérac | 1965–2010 |
| Larra | 1965–2010 | Loubéjac | 1962–2010 | Tonneins | 1962–2010 |

Climatic data consist of the 8-km SAFRAN product [46,47] from the French weather agency (Météo-France): a reanalysis based on their ARPEGE model [48]. It extends from 1958 to 2014, overlaying the available data from hydrological gauging stations.

Projected climatic data originate from 10 regional climatic models (RCM) forced at their boundaries by a global climatic model (GCM). Each GCM/RCM pair produce the 1972–2010 climate as well as the projected climate over 2010–2100 using RCP 4.5 and/or RCP 8.5 emission scenarios [49]. A total of 10 historical runs (one per pair) and 16 future runs (regarding RCP scenarios) are available for the study (Table 3). Climate projections have been provided by DRIAS services (http://www.drias-climat.meteo.fr): a Météo-France (CNRM), Pierre-Simon-Laplace Institute

(IPSL) and European center for research and advanced teaching in scientific calculation (CERFACS) collaboration. GCM/RCM pairs are from the EURO-CORDEX project [50], corrected against SAFRAN data by DRIAS services. More specifically, the CNRM-ARPEGE_CNRM-ALADIN data have been corrected using a quantile-quantile method while the others have been locally adjusted through a cumulative distribution function transform (CDFt). Available data have also been downscaled from the native EURO-CORDEX grid (~12 km) to the SAFRAN grid (8 km). The latter is of great interest to our study, avoiding possible biases induced by a change in the spatial positioning of the weather data provided to the SWAT model between calibration and projection. Daily precipitation, as well as minimum and maximum temperature, were used in this study.

**Table 3.** Climatic models available for the study.

| GCM | RCM | SCENARIOS | | |
| --- | --- | --- | --- | --- |
| | | Historical | RCP 4.5 | RCP 8.5 |
| CNRM-ARPEGE | CNRM-ALADIN | X | X | X |
| CNRM-CERFACS-M5 | RC4A | X | X | X |
| CNRM-CM5 | CCLM4-8-17 | X | X | |
| EC-EARTH | RCA4 | X | X | X |
| IPSL-CM5A-MR | RCA4 | X | | X |
| IPSL-CM5A-MR | WRF331F | X | X | |
| MPI-ESM-LR | CCLM4-8-17 | X | X | X |
| MPI-ESM-LR | REMO019 | X | X | X |
| MPI-M-MPI-ESM-LR | RCA4 | X | | X |
| MetEir-EC-EARTH | RACMO22E | X | X | X |

The discretization of the watershed into sub-watersheds aimed a fair representation of the hydrological processes and reasonable computing time allocation. The process led to the delineation of 150 subbasins, the outlet of some of them matching one of the 21 gauging stations. HRUs are defined on soil, land use, and slope, retaining only information covering more than 10% of the subbasin area to eliminate anecdotal HRUs as proposed by Srinivasan [51]. Elevation bands are introduced in mountainous area, as described in Grusson, et al. [52], to enhance snow pack simulation. As only precipitation and temperature data were available from climate models, SWAT relied on the potential evapotranspiration formulation proposed by Hargreaves [40].

*2.3. Model Calibration and Validation*

SWAT-CUP [53] is used to identify sensitive parameters in the SWAT model, as well as for calibration, using the SUFI-2 algorithm [54]. SWAT-CUP is a tool that allows SWAT users to perform automatic calibrations [55]. Among the different calibration algorithms offered by SWAT-CUP to perform the calibration, SUFI-2 is known to identify an appropriate parameter set in a limited number of iterations [56]. The sensitivity analysis has been conducted through a one-at-a-time procedure [53]. Thirty-two parameters were considered for this analysis (Table 4). Five runs were performed for each parameters over the 10-year period from 2000 to 2010, preceded by a 3-year warming period (1997–2000). At the end of this process, a set of sensitive parameters has been selected for each of the 21 gauging stations in order to calibrate their upstream subwatersheds.

Calibration step consisted of a 1500 runs proceeding, as recommended by Yang, et al. [56]. SWAT-CUP calibration was achieved sequentially, upstream to downstream, one gauging station at the time, based on the Nash-Sutcliffe efficiency criterion (NSE) [57]. NSE is a normalized metric comparing the variance of the observed dataset and the variance of the observation and simulation residual errors. NSE ranges from $-\infty$ to 1 and is sensitive to large errors. It equals 0 when the model is as accurate as the mean of the observations, and equals 1 when the model achieves a perfect fit.

**Table 4.** Parameters considered for the sensitivity analysis.

| Parameters | Description | Min | Max | Default |
|---|---|---|---|---|
| **HYDOLOGICAL PARAMETERS** | | | | |
| EPCO | Plant uptake compensation factor | 1 | 0 | 1 |
| SURLAG | Surface runoff lag time | 0.5 | 1 | 4 |
| GW_Delay | Groundwater delay | 0 | 500 | 31 |
| GW_Revap | Groundwater "revap" coefficient. | 0.02 | 0.2 | 0.02 |
| GWQMN | Threshold in the shallow aquifer for return flow to occur | 0 | 5000 | 1000 |
| GWHT | Initial groundwater height | 0 | 25 | 1 |
| GW_SPYLD | Specific yield of the shallow aquifer | 0 | 0.4 | 0.003 |
| SHALLST | Initial depth of water in the shallow aquifer | 0 | 50000 | 500 |
| DEEPST | Initial depth of water in the deep aquifer | 0 | 50000 | 1000 |
| ALPHA_BF | Base flow alpha factor (days) | 0 | 1 | 0.048 |
| REVAPMN | Threshold in the shallow aquifer for 'revap' to occur | 0 | 500 | 0 |
| RCHRG_DP | Deep aquifer percolation fraction | 0 | 1 | 0.05 |
| ESCO | Soil evaporation compensation factor | 0 | 1 | 0.95 |
| CN2 (relative test) | SCS runoff curve number | −0.2 | 0.2 | HRU |
| CANMX | Maximum canopy storage | 0 | 100 | HRU |
| OV_N | Manning's "n" value for overland flow | 0.01 | 30 | HRU |
| SOL_AWC (relative test) | Available water capacity of the soil layer | −0.5 | 0.5 | soil layer |
| SOL_K (relative test) | Saturated hydraulic conductivity | −10 | 10 | soil layer |
| SOL_Z (relative test) | Depth from soil surface to bottom of layer | −500 | 500 | soil layer |
| EVRCH | Reach evaporation adjustment factor | 0.5 | 1 | 1 |
| EVLAI | LAI at which no evaporation occurs from water surface | 0 | 10 | 3 |
| **SNOW PARAMETERS** | | | | |
| SFTMP | Snowfall temperature | −10 | 10 | 4.5 |
| SMTMP | Snowmelt base temperature | −10 | 10 | 4.5 |
| TIMP | Snowpack temperature lag factor | 0 | 1 | 1 |
| SMFMX | Maximum melt rate for snow during year (summer solstice) | 0 | 20 | 1 |
| SMFMN | Minimum melt rate for snow during year (winter solstice) | 0 | 20 | 0.5 |
| SNOW50COV | Snow water equivalent that corresponds to 50% snow cover | 0 | 1 | 0.5 |
| SNOWCOVMX | Snow water content that corresponds to 100% snow cover | 0 | 100 | 1 |
| SNO_SUB | Initial snow water content | 0 | 300 | 0 |
| **ELEVATION BAND PARAMETERS** | | | | |
| TLAPS | Temperature lapse rate | −10 | 10 | −6 |
| PLAPS | Precipitation lapse rate | −100 | 500 | 0 |
| SNOEB | Initial snow water content in elevation bands | 0 | 300 | 0 |

Performance is afterward evaluated on several criterion including NSE, but also with the NSE calculated on the root square of the discharge values (NSEsqrt) and on the logarithm of the discharge values (NSElog). NSEsqrt reduces the influence of large errors on the metric, while NSElog provides more weight on low flows [58–60]. The percent bias (Pbias) is calculated as well. It informs on the deviation between simulated and observed values, ranging from 0 to ±∞, positive values indicating overestimation and negative values indicating underestimation. Finally, the coefficient of determination ($R^2$), extending from −1 to 1, is also computed. Despite known limitations of this metrics to evaluate simulation performance [61], it is still widely used by scientific community, and has been therefore added to complete the array of metrics presented in this study.

For each station, the 2000–2010 period is used for calibration and all remaining data have then be use for validation: out of the 21 stations, eleven have been validated over a 38-year period (1962–1999), and the shorter validation period for the Truyère-Aval station, covers 21 years. Validation has been conducted over the longest available period for each station, details can be found in Table 2. The calibrated model are used to realize several runs: one run with SAFRAN data over the historical period (1962–2010), 10 runs using data from climate models over the historical period (1972–2010) and 16 runs using projected climate data over 2010–2050 (8 GCM/RCM pairs per scenarios, see Table 3).

*2.4. Ability of Climate Models to Represent Our Regional Climate*

As a first step, the 10 runs performed with historical data from climate models are used to evaluate against SAFRAN the ability of climate models to depict our regional climate. During this step, the behaviors of the calibrated SWAT model to simulate the discharge when feed with such dataset is

also evaluate. The distribution of temperature and precipitation for both dataset as well as simulated and observed discharge at the outlet of the watershed are compared over the 1972–2010 period.

*2.5. Change within the Water Cycle*

In a second step, change within hydrological cycle are investigated for the most relevant hydrological components simulated by the SWAT model: discharge, snowpack, surface runoff, infiltration, subsurface flow, evapotranspiration, and soil water content.

Monthly decadal seasonal averages are also calculated for each subwatershed from the run using SAFRAN over the past period. Two 10-year periods are considered, namely 1962–1972 and 2000–2010, which span over about 50 years. Both decade periods are compared through the following value, describing the evolution over time

$$\Delta = \text{Volume}_{2000–2010} - \text{Volume}_{1962–1972} \tag{1}$$

This analysis allows to assess the changes from a spatial perspective, and better understand the water cycle functioning within the watershed, helping to better understand and interpret the changes highlighted over the overall century period covered by the study.

Monthly seasonal data for the watershed are also calculated from the entire time series, for winter (December, January, February; DJF) and summer (June, July, August; JJA), from historical simulation using SAFRAN and projected simulation using climate model dataset. For each season and each run, a Mann–Kendall test [62,63], as modified by Hirsch, et al. [64], is performed to identify significant trends over the period. The Theil–Sen estimator provides additional trend information on the intensity of the variation. It is a non-parametric approach that robustly adjusts a line to a series using the median of the slopes of all lines connecting each pairs of the series [65,66].

## 3. Results

*3.1. Model Performance: Calibration and Validation*

Figure 2 details the performance of discharge simulations over calibration and validation periods, at each of the 21 gauging stations. Most of performance values are deemed 'very good' to 'satisfactory', based on the ranking proposed by Moriasi, et al. [67] for monthly time step verification.

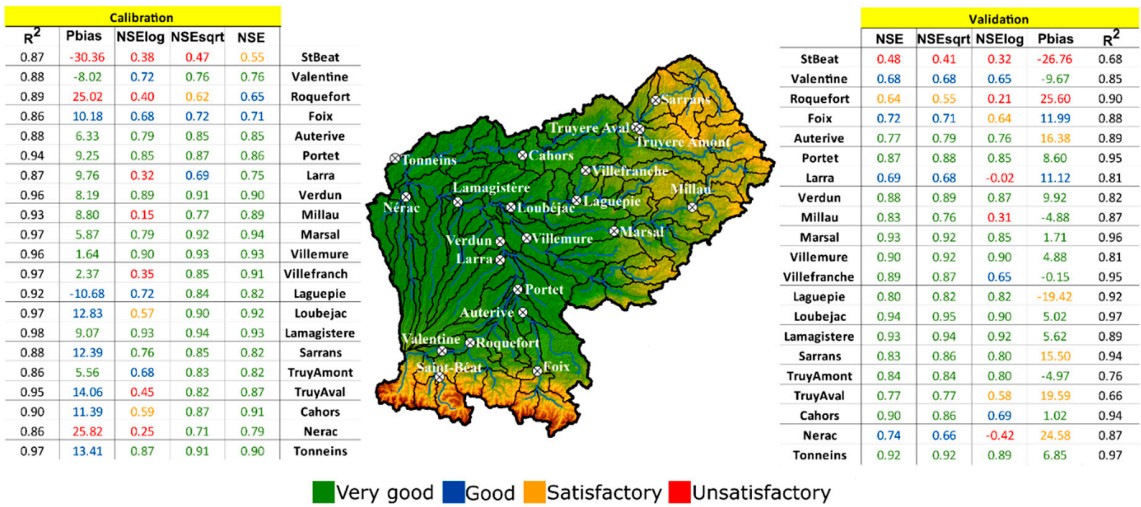

**Figure 2.** Performances in calibration and validation (following Moriasi, et al. [67]).

In calibration, only three sites (NSE values) fall in the category 'satisfactory' (St. Beat) or 'good' (Roquefort and Foix). All 18 other gauging stations attain 'very good' NSE values. St. Beat is in fact the only site for which all performance values range from 'satisfactory' to 'unsatisfactory'. Roquefort does

better, oscillating between 'good' and 'unsatisfactory', when Foix always displays 'good' performance values. $R^2$ is excluded from the ranking proposed by Moriasi, et al. [67], but its values here are always superior to 0.86. Most of the 'unsatisfactory' performance values are obtained for the NSElog, highlighting the difficulties of the model to simulated low flows at seven of the gauging stations. Validation performance values offer a similar picture than calibration one. Poorer performance is again achieved at St. Beat, Roquefort, and Foix than at any other sites, while Larra and Nerac are the only two sites with NSE decreasing from 'very good' to 'good'. Difficulties of the SWAT model to simulate some low flows persist but does not worsen.

### 3.2. Representativeness of Climatic Ensemble and Hydro-Climatic Chain

Prior to exploring the future hydrologic regime of the Garonne watershed, climatic runs issued by the 10 available GCM/RCM pairs are evaluated against the SAFRAN dataset for the 1972–2010 historical period, as illustrated in Figure 3 in which yearly precipitation and average temperature are compared. Cumulative distribution function curves in this figure confirm that SAFRAN lies well within the GCM/RCM ensemble. The precipitation from climatic ensemble overshoots extreme SAFRAN values: from 0% to 10% and upper than 90% of the distribution. The same conclusion can be drawn from the cumulative distribution function curve of temperature.

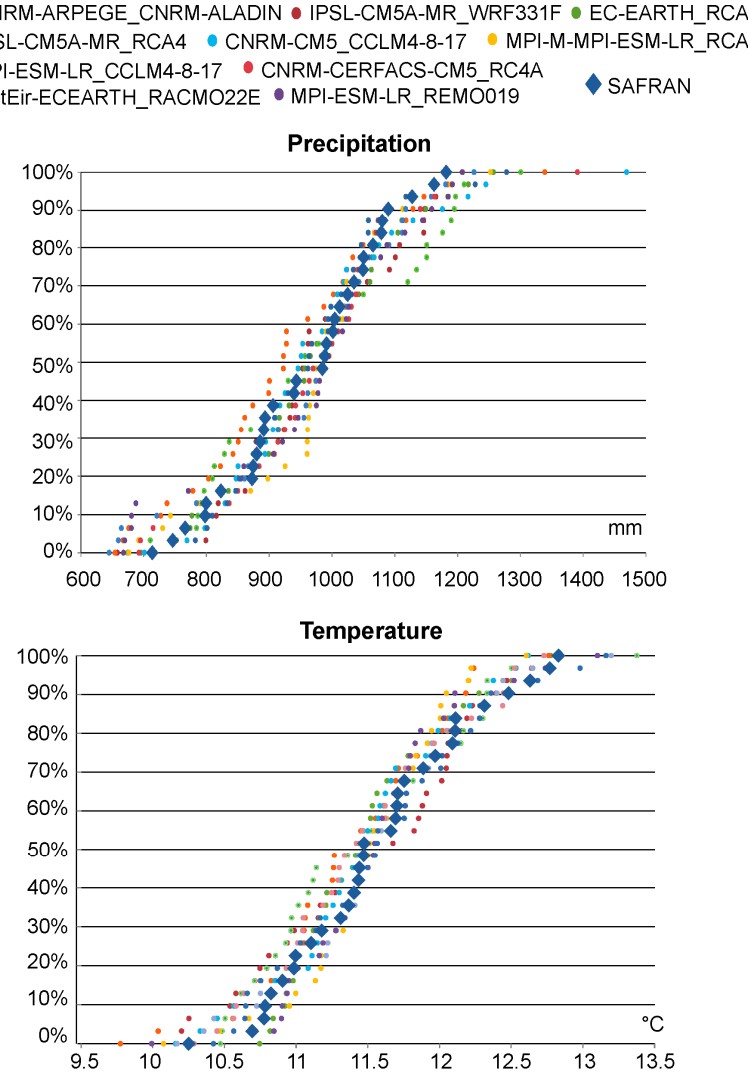

**Figure 3.** Comparison of yearly precipitation and average temperature climatic ensembles (historical period) against the SAFRAN dataset. (Names as <GCM-name_RCM-name>).

Figure 4 compares the average annual discharge at Tonneins simulated from each of the dataset over the historical period, as well as in situ measurements. Distribution of the SAFRAN and the observed discharge series are quite similar as expected from the good results of the calibration procedure. Both of them fall into the range described by CDF curves obtained from the climate ensemble. It nevertheless can be notice that extreme high discharge values (upper 10%), are overestimated by the hydro-climatic chain simulations and shows a larger dispersion than for the rest of the distribution. This is consistent with the overestimation of precipitation for this part of the distribution observed above.

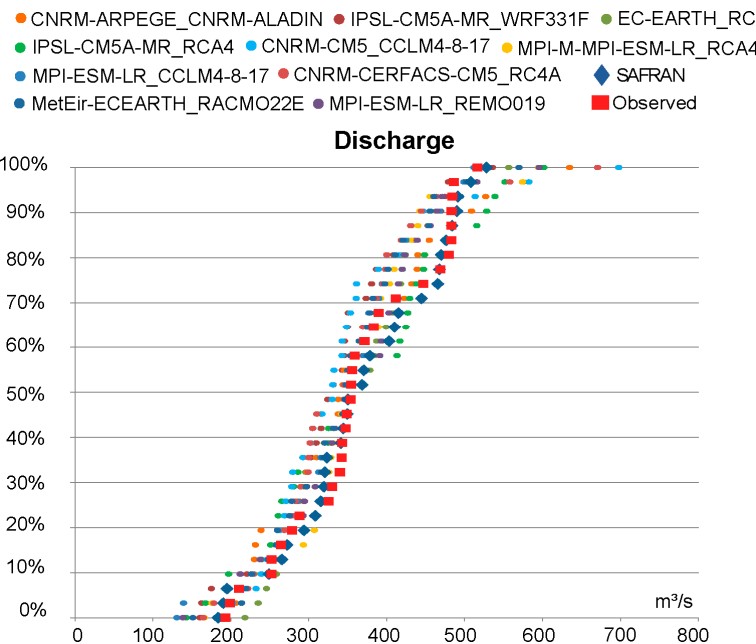

**Figure 4.** Comparison of discharge simulated from each dataset: climatic ensembles (historical period) and the SAFRAN dataset. (Names as <GCM-name_RCM-name>).

### 3.3. Change within the Water Cycle

The Table 5 and Figure 5 depict the evolution of temperatures and precipitations. A substantial difference is visible between winter and summer. During winter, the temperature is found increasing for the majority of models but this increase is significant only over the projected period. No clear trend emerges from the analysis of winter precipitation, whether historical or projected periods. Some series indicate a decrease when others indicate an increase but very few of them are shown as significant by Mann–Kendall tests. Trends are more significant when considering summer. An increase of temperature and a decrease of precipitation are expected during this season for the 2010–2050 period, with significant $p$-values for the majority of climate models. Those trends are in accordance with the trends observed over the historical period. It is interesting here to notice that trends are globally in accordance when comparing the past and the future period and also that trends from the SAFRAN dataset are consistent with the climate ensemble over the historical period.

**Table 5.** Trend in precipitation and temperature over winter (DJF) and summer (JJA) seasons at watershed scale, for SAFRAN (historical period) and the climatic ensemble (historical and projected period). Bold numbers are significant trend considering $\alpha$ = 0.1 and underlined numbers considering $\alpha$ = 0.05.

| | DJF | | | | | | JJA | | | | | |
|---|---|---|---|---|---|---|---|---|---|---|---|---|
| | Temperature | | | Precipitation | | | Temperature | | | Precipitation | | |
| | T (trend) | *P*-Values | Sen's Slope | T (trend) | *P*-Values | Sen's Slope | T (trend) | *P*-Values | SEN'S slope | T (trend) | *P*-Values | Sen's Slope |
| **Historical period (1972–2010)** | | | | | | | | | | | | |
| **SAFRAN (observed)** | + | 0.3 | 0.01 | - | 0.63 | −0.11 | + | <u><0.001</u> | 0.06 | - | **0.02** | −3.21 |
| CNRM-ARPEGE_CNRM-ALADIN | + | 0.274 | 0.02 | - | 0.42 | −0.01 | + | <u>0.021</u> | 0.04 | - | <u><0.001</u> | −1.31 |
| CNRM-CERFACS-CM5_RC4A | + | 0.701 | 0.01 | + | 0.74 | 0.24 | + | **0.078** | 0.04 | - | <u>0.002</u> | −3.72 |
| CNRM-CM5_CCLM4-8-17 | - | 0.854 | −0.01 | + | 0.27 | 0.76 | + | 0.652 | 0.01 | - | <u><0.001</u> | −17.67 |
| ICHEC-EARTH_RCA4 | + | 0.469 | 0.01 | - | 0.65 | −0.34 | + | 0.463 | 0.01 | - | <u>0.03</u> | −8.92 |
| IPSL-CM5A-MR_RCA4 | + | 0.143 | 0.03 | - | 0.34 | −1.60 | + | 0.309 | 0.03 | - | <u>0.003</u> | −12.83 |
| IPSL-CM5A-MR_WRF331F | + | 0.16 | 0.04 | - | 0.18 | −1.71 | + | **0.078** | 0.04 | - | 0.59 | −2.47 |
| MPI-ESM-LR_CCLM4-8-17 | + | 0.541 | 0.01 | + | 0.59 | −0.10 | + | **0.069** | 0.04 | - | 0.15 | −5.70 |
| MPI-ESM-LR_REMO019 | + | 0.876 | 0 | - | 0.75 | −1.33 | + | 0.283 | 0.02 | - | 0.65 | −1.26 |
| MPI-M-MPI-ESM-LR_RCA4 | + | 0.371 | 0.02 | + | 0.73 | 0.32 | + | 0.174 | 0.03 | - | 0.35 | −2.44 |
| MetEir-ECEARTH_RACMO22E | + | <u>0.039</u> | 0.04 | - | 0.39 | −0.67 | + | <u>0.043</u> | 0.05 | - | 0.83 | 1.14 |
| **Projection (2010–2050)—RCP 4.5** | | | | | | | | | | | | |
| CNRM-ARPEGE_CNRM-ALADIN | + | <u>0.029</u> | 0.09 | + | 0.69 | 0.64 | + | 0.34 | 0.01 | - | <u><0.001</u> | −25.21 |
| CNRM-CERFACS-CM5_RC4A | + | **0.054** | 0.07 | - | 0.56 | −0.59 | + | 0.25 | 0.03 | - | **0.098** | −6.34 |
| CNRM-CM5_CCLM4-8-17 | + | <u>0.036</u> | 0.09 | - | 0.14 | −1.25 | + | <u>0.001</u> | 0.06 | - | <u>0.001</u> | −12.71 |
| ICHEC-EARTH_RCA4 | + | <u>0.012</u> | 0.09 | + | **0.09** | 1.83 | + | <u>0.02</u> | 0.02 | - | <u>0.035</u> | −6.11 |
| IPSL-CM5A-MR_WRF331F | + | **0.052** | 0.05 | + | 0.63 | 0.63 | + | <u>0.04</u> | 0.01 | - | 0.434 | −2.89 |
| MPI-ESM-LR_CCLM4-8-17 | + | **0.073** | 0.05 | + | 0.71 | −0.02 | + | <u>0.03</u> | 0.03 | - | <u><0.001</u> | −11.90 |
| MPI-ESM-LR_REMO019 | + | 0.185 | 0.02 | - | 0.90 | −0.22 | + | <u>0.04</u> | 0.03 | - | <u><0.001</u> | −11.70 |
| MetEir-ECEARTH_RACMO22E | + | **0.081** | 0.05 | - | 0.54 | −1.20 | + | <u>0.01</u> | 0.03 | - | <u>0.042</u> | −5.22 |
| **Projection (2010–2050)—RCP 8.5** | | | | | | | | | | | | |
| CNRM-ARPEGE_CNRM-ALADIN | + | <u>0.001</u> | 0.14 | + | 0.99 | −0.16 | + | <u><0.001</u> | 0.07 | - | <u><0.001</u> | −26.95 |
| CNRM-CERFACS-CM5_RC4A | + | <u>0.01</u> | 0.09 | + | 0.44 | 0.25 | + | 0.62 | 0.01 | - | <u>0.04</u> | −13.32 |
| ICHEC-EARTH_RCA4 | + | <u>0.001</u> | 0.07 | - | 0.81 | 0.18 | + | <u><0.001</u> | 0.07 | − | <u>0.002</u> | −11.12 |
| IPSL-CM5A-MR_RCA4 | + | 0.37 | 0.01 | + | <u>0.02</u> | 2.14 | + | <u>0.01</u> | 0.05 | - | 0.14 | −0.99 |
| MPI-ESM-LR_CCLM4-8-17 | + | **0.07** | 0.06 | + | 0.19 | 1.23 | + | <u>0.003</u> | 0.04 | - | <u>0.02</u> | −9.32 |
| MPI-ESM-LR_REMO019 | + | 0.31 | 0.03 | + | **0.10** | 1.18 | + | <u>0.02</u> | 0.02 | - | <u>0.02</u> | −4.28 |
| MPI-M-MPI-ESM-LR_RCA4 | + | **0.06** | 0.03 | + | 0.13 | 1.23 | + | <u>0.02</u> | 0.04 | - | <u>0.003</u> | −8.92 |
| MetEir-ECEARTH_RACMO22E | + | <u>0.02</u> | 0.06 | + | 0.54 | 0.32 | + | <u>0.02</u> | 0.03 | - | 0.16 | −5.06 |

Figure 5 present more information about the geographic spread of those evolutions over the historical period. Consistently with the Mann–Kendall test, winter seems balance between small increase and small decrease distributed over the watershed. The most noticeable increase of precipitation took place in the Pyrenees. On the other hand, during the summer, if some few watersheds have experienced a slight increase of precipitation, most of it have been impacted by a deficit of seasonal precipitation compared to the 1962–1972 period. This decrease seems also more important over mountainous areas.

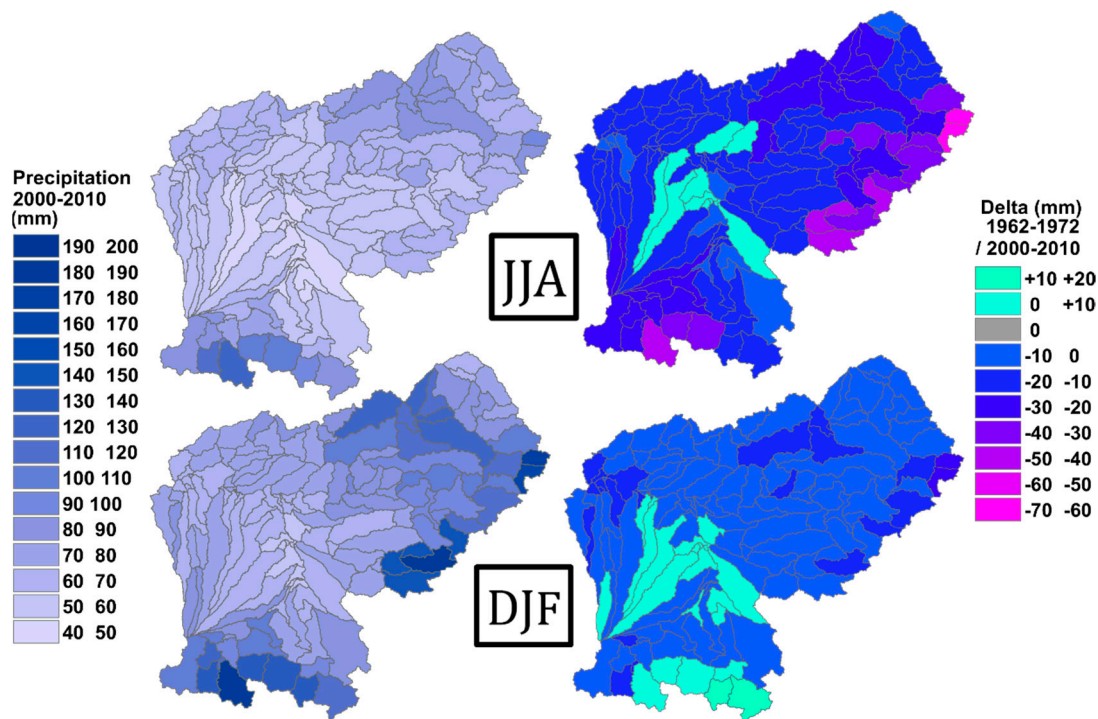

**Figure 5.** Total Precipitation during the 2000–2010 decade and variations with the 1962–1972 decade (seasonal mean monthly values).

Snowpack, infiltration, runoff, subsurface flow, evapotranspiration, soil water content, and discharge evolutions have also been compared, for winter (DJF) and summer (JJA). Surface blue water components (discharge, snowpack, surface runoff, infiltration, and subsurface flow) are shown in Tables 6–10 and Figures 6–10 and Green water component, namely evapotranspiration and soil water, content are shows in Tables 11 and 12 and Figures 11 and 12. As for climate evolutions, the projection of hydrological components is analyzed together with the changes simulated from the SAFRAN dataset over the last 50 years in order to better understand mechanisms taking place over the watershed.

According to the Mann–Kendall test, discharge at the outlet of the watershed has significantly decreased during the last 50 years for both seasons. This decrease of discharge has taken place almost everywhere over the watershed except for few reaches in the plain during summer and in the mountain during winter (Figure 6). This tendency is expected to be the same over the coming 30 years (Table 6). All trends obtained from the climate ensemble are highly homogenous: all models for each scenario/season show a decrease of discharge. This trend appears however more significant during the summer season compared to the winter.

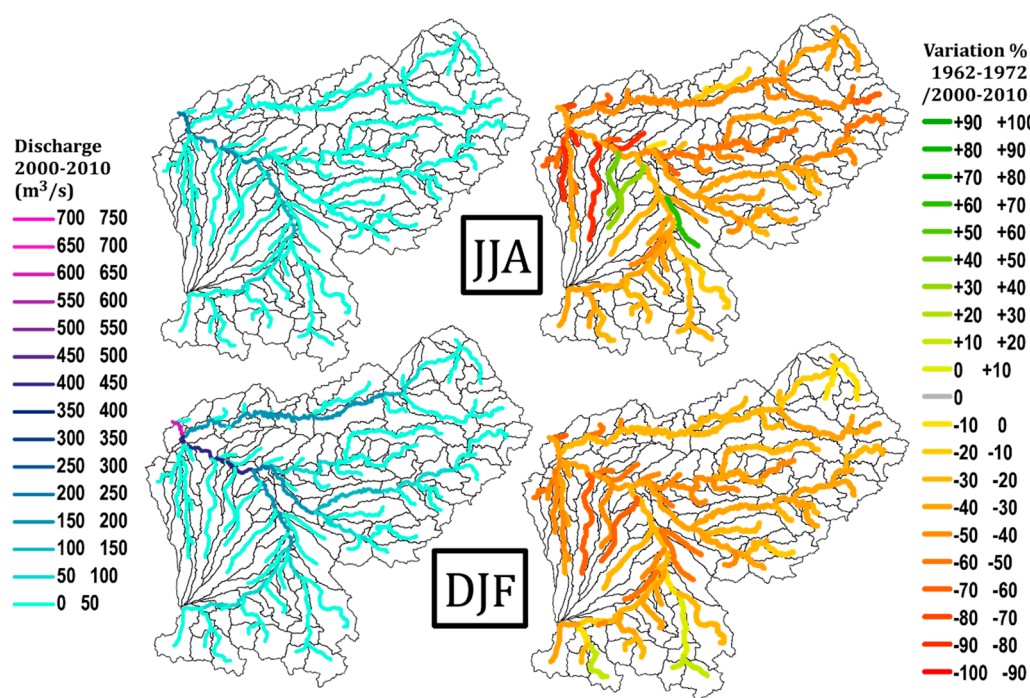

**Figure 6.** Discharge during the 2000–2010 decade and variations with the 1962–1972 decade.

**Table 6.** Trend in discharge over winter (DJF) and summer (JJA) seasons at watershed scale, for SAFRAN (historical period) and climatic ensemble (projected period). Bold numbers are significant trend considering α = 0.1 and underlined numbers considering α = 0.05.

| | Discharge | | | | | |
|---|---|---|---|---|---|---|
| | **DJF** | | | **JJA** | | |
| | **T (trend)** | **P-Value** | **Sen's Slope** | **T (trend)** | **P-Value** | **Sen's Slope** |
| **Historical Period (1962–2010)** | | | | | | |
| SAFRAN | - | **0.031** | −4.99 | - | **<0.001** | −12.56 |
| **Project period (2010–2050)—RCP 4.5** | | | | | | |
| CNRM-ARPEGE_CNRM-ALADIN | - | **0.02** | −2.01 | - | **<0.001** | −1.78 |
| CNRM-CERFACS-CNRM-CM5_RCA4 | - | 0.11 | −1.42 | - | **<0.001** | −1.95 |
| CNRM-CM5_CCLM4-8-17 | - | **0.005** | −2.11 | - | **<0.001** | −1.89 |
| ICHEC-EC-EARTH_RCA4 | - | **0.05** | −1.22 | - | **<0.001** | −1.05 |
| IPSL-IPSL-CM5A-MR_WRF331F | - | **0.03** | −1.68 | - | **<0.001** | −1.28 |
| MPI-ESM-LR_CCLM4-8-17 | - | 0.16 | −1.37 | - | **<0.001** | −1.12 |
| MPI-ESM-LR_REMO019 | - | 0.11 | −1.55 | - | **<0.001** | −1.98 |
| MetEir-ECEARTH_RACMO22E | - | **0.01** | −2.48 | - | **<0.001** | −1.32 |
| **Project period (2010–2050)—RCP 8.5** | | | | | | |
| CNRM-ARPEGE_CNRM-ALADIN | - | **0.02** | −2.00 | - | **<0.001** | −2.08 |
| CNRM-CERFACS-CNRM-CM5_RCA4 | - | 0.65 | −1.02 | - | **<0.001** | −2.23 |
| ICHEC-EC-EARTH_RCA4 | - | **0.03** | −1.95 | - | **<0.001** | −2.86 |
| IPSL-IPSL-CM5A-MR_RCA4 | - | 0.54 | −0.36 | - | **<0.001** | −6.56 |
| MPI-ESM-LR_CCLM4-8-17 | - | 0.57 | −0.78 | - | **<0.001** | −1.89 |
| MPI-ESM-LR_REMO019 | - | 0.18 | −7.59 | - | **<0.001** | −2.42 |
| MPI-M-MPI-ESM-LR_RCA4 | - | 0.65 | −3.55 | - | **<0.001** | −1.76 |
| MetEir-ECEARTH_RACMO22E | - | 0.44 | −1.18 | - | **<0.001** | −1.93 |

The increase of discharge in mountains (up to +10%) for the historical period is following a reduction of the snowpack height from −20 to −50%, as visible on Figure 7. This reduction of snow pack simulated from the SAFRAN dataset is significant, and this trend is projected to remain the same in the future as show in Table 7 where a significant decrease is also given by all climate models but one. A balance of increase and decreasing trend is visible within subwatershed in the plain, but snow is rare in those areas and delta values are extremely low with values under 1 mm for most of them.

**Table 7.** Trend in snow stock over winter (DJF) season at watershed scale, for SAFRAN (historical period) and climatic ensemble (projected period). Bold numbers are significant trend considering α = 0.1 and underlined numbers considering α = 0.05.

| | Snow Stock | | |
| --- | --- | --- | --- |
| | DJF | | |
| | T (trend) | *P*-Value | Sen's Slope |
| Historical Period (1962–2010) | | | |
| SAFRAN | - | **<0.001** | −0.17 |
| Project period (2010–2050)—RCP 4.5 | | | |
| CNRM-ARPEGE_CNRM-ALADIN | - | **<0.001** | −0.38 |
| CNRM-CERFACS-CNRM-CM5_RCA4 | - | **0.003** | −0.16 |
| CNRM-CM5_CCLM4-8-17 | - | **0.01** | −0.21 |
| ICHEC-EC-EARTH_RCA4 | - | **0.004** | −0.14 |
| IPSL-IPSL-CM5A-MR_WRF331F | - | **0.02** | −0.13 |
| MPI-ESM-LR_CCLM4-8-17 | - | **0.002** | −0.22 |
| MPI-ESM-LR_REMO019 | - | **0.01** | −0.19 |
| MetEir-ECEARTH_RACMO22E | - | **0.001** | −0.25 |
| Project period (2010–2050)—RCP 8.5 | | | |
| CNRM-ARPEGE_CNRM-ALADIN | - | **0.004** | −0.29 |
| CNRM-CERFACS-CNRM-CM5_RCA4 | - | **0.01** | −0.12 |
| ICHEC-EC-EARTH_RCA4 | - | **0.04** | −0.05 |
| IPSL-IPSL-CM5A-MR_RCA4 | - | **0.10** | −0.09 |
| MPI-ESM-LR_CCLM4-8-17 | - | **0.03** | −0.10 |
| MPI-ESM-LR_REMO019 | - | 0.22 | −0.03 |
| MPI-M-MPI-ESM-LR_RCA4 | - | **0.02** | −0.05 |
| MetEir-ECEARTH_RACMO22E | - | **0.02** | −0.21 |

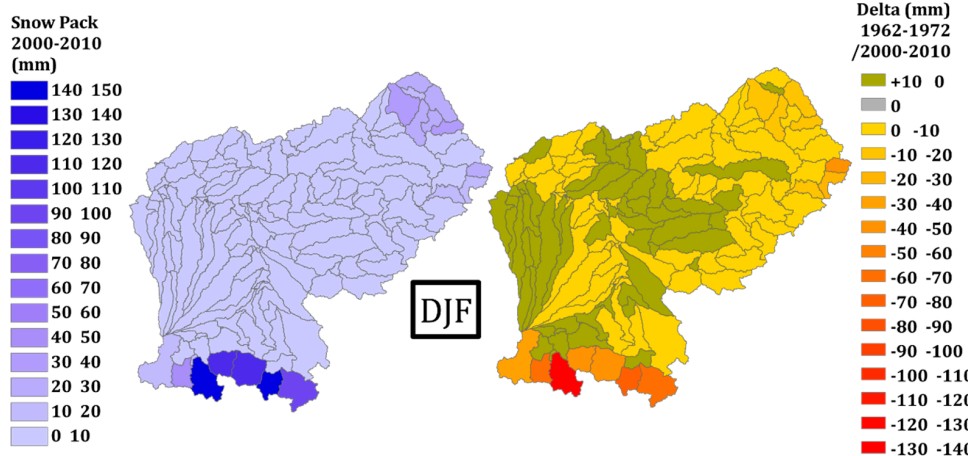

**Figure 7.** Snowpack during the 2000–2010 decade and variations with the 1962–1972 decade.

The variation of surface run off is also significant over the historical period, with a decrease in winter and summer (Table 8). If looking at Figure 8 to understand the geographic distribution during the last 50 years, the winter decrease was global, with the exception of the mountainous subwatershed where rain is increasing and snowpack decreasing. The variation over summer seems somehow limited to less than 2 mm and balance geographically, the negative trend over the overall period being triggered by some few subwatershed with a higher decrease of run off mainly situated in the northeast part of the basin. When looking at the projections, the summer season appear rather stable for the next 30 years. Over this period, and for the RCP 4.5 scenario, half of climate models indicate an increase when half indicate a decrease. Only one decreasing trend appear significant. For RCP 8.5 scenario, a majority of models indicates an increase, but the *p*-values are quite high indicating a very poor significance. For the summer period on the other hand, a general and significant decrease of surface run off is highlighted by the statistical tests, in accordance to the observation made over the historical period.

**Table 8.** Trend in surface run off over winter (DJF) and summer (JJA) seasons at watershed scale, for SAFRAN (historical period) and climatic ensemble (projected period). Bold numbers are significant trend considering $\alpha = 0.1$ and underlined numbers considering $\alpha = 0.05$.

| | Surface run off | | | | | |
|---|---|---|---|---|---|---|
| | DJF | | | JJA | | |
| | T (trend) | *P*-Value | Sen's Slope | T (trend) | *P*-Value | Sen's Slope |
| Historical Period (1962–2010) | | | | | | |
| SAFRAN | - | **0.01** | −0.08 | - | 0.1 | −0.02 |
| Project period (2010-2050)—RCP 4.5 | | | | | | |
| CNRM-ARPEGE_CNRM-ALADIN | + | 0.74 | 0.08 | - | **<0.001** | −1.75 |
| CNRM-CERFACS-CNRM-CM5_RCA4 | - | 0.26 | −0.17 | - | **0.01** | −0.54 |
| CNRM-CM5_CCLM4-8-17 | - | **0.01** | −0.39 | - | **<0.001** | −1.01 |
| ICHEC-EC-EARTH_RCA4 | + | 0.21 | 0.25 | - | **0.02** | −0.40 |
| IPSL-IPSL-CM5A-MR_WRF331F | + | 0.98 | 0.03 | - | **0.02** | −0.42 |
| MPI-ESM-LR_CCLM4-8-17 | + | 0.73 | 0.03 | - | **<0.001** | −0.88 |
| MPI-ESM-LR_REMO019 | - | 0.88 | −0.02 | - | **<0.001** | −0.90 |
| MetEir-ECEARTH_RACMO22E | - | 0.12 | −0.30 | - | **0.004** | −0.38 |
| Project period (2010–2050)—RCP 8.5 | | | | | | |
| CNRM-ARPEGE_CNRM-ALADIN | - | 0.74 | −0.07 | - | **<0.001** | −1.84 |
| CNRM-CERFACS-CNRM-CM5_RCA4 | + | 0.70 | −0.01 | - | **0.01** | −1.16 |
| ICHEC-EC-EARTH_RCA4 | - | 0.49 | −0.05 | - | **<0.001** | −0.90 |
| IPSL-IPSL-CM5A-MR_RCA4 | + | 0.12 | 0.30 | - | **0.004** | −0.21 |
| MPI-ESM-LR_CCLM4-8-17 | + | 0.43 | 0.16 | - | **<0.001** | −0.85 |
| MPI-ESM-LR_REMO019 | + | 0.98 | −0.01 | - | **0.003** | −0.77 |
| MPI-M-MPI-ESM-LR_RCA4 | + | 0.44 | 0.14 | - | **<0.001** | −0.81 |
| MetEir-ECEARTH_RACMO22E | + | 0.78 | −0.02 | - | **0.001** | −0.46 |

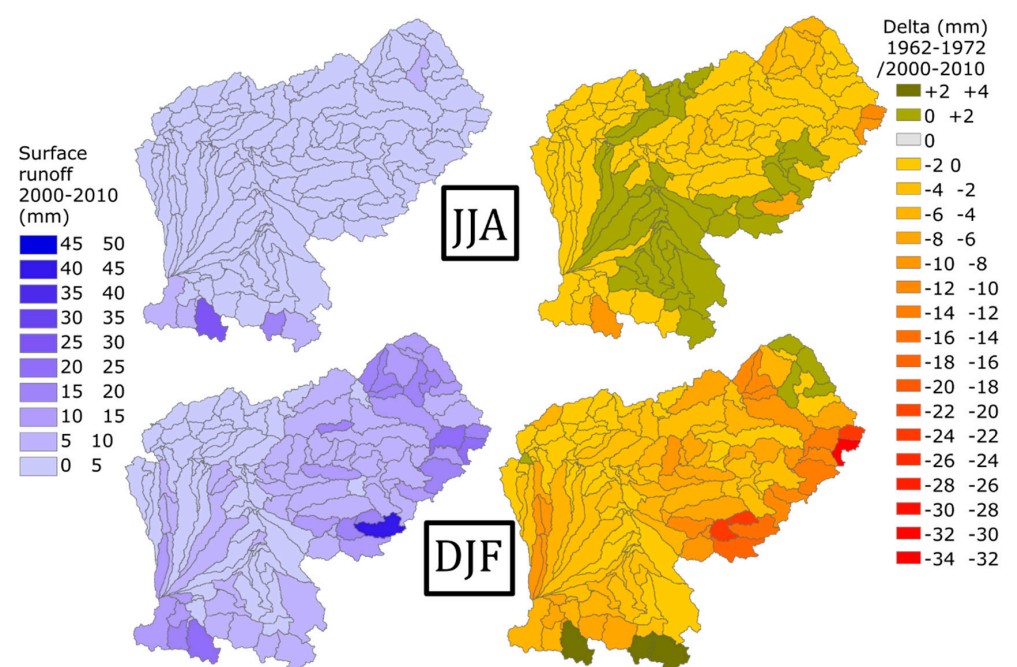

**Figure 8.** Runoff during the 2000–2010 decade and variations with the 1962–1972 decade.

The flow of infiltrated water appears also to have decreased during the last 50 years, but winter is the only season where this trend is found significant. If looking at Figure 9 we can see that the geographic spread of the winter decreasing trend over the last 50 years was global, with higher rate in mountainous area. During the summer, most of the watershed has also been impacted by a decrease with some exception, notably in the Pyrenean part, presenting substantial values of a positive delta. As for run off, the decreasing trend for winter does not appear clearly when considering climate projections (Table 9). For this season, the RCP 4.5 scenario shows no significant trend and the ratio increase/decrease is balanced among the ensemble. With the RCP 8.5 scenario, even if this ratio is also balanced, three decreasing trends appears however significant when increasing trend does not. For summer season, the decreasing trend is consistent over models and scenarios, with a majority

of significant trend, which seems to indicate an intensification of this trends already visible over the historical period.

**Table 9.** Trend in infiltration over winter (DJF) and summer (JJA) seasons at watershed scale, for SAFRAN (historical period) and climatic ensemble (projected period). Bold numbers are significant trend considering $\alpha = 0.1$ and underlined numbers considering $\alpha = 0.05$.

| | Infiltration | | | | | |
| --- | --- | --- | --- | --- | --- | --- |
| | DJF | | | JJA | | |
| | T (trend) | *P*-Value | Sen's Slope | T (trend) | *P*-Value | Sen's Slope |
| Historical Period (1962–2010) | | | | | | |
| SAFRAN | - | **0.05** | −0.26 | - | 0.54 | −0.05 |
| Project period (2010–2050)—RCP 4.5 | | | | | | |
| CNRM-ARPEGE_CNRM-ALADIN | + | 0.71 | 0.54 | - | <u>**0.003**</u> | −9.53 |
| CNRM-CERFACS-CNRM-CM5_RCA4 | - | 0.56 | −0.49 | - | **0.06** | −11.29 |
| CNRM-CM5_CCLM4-8-17 | - | 0.23 | −0.82 | - | 0.18 | −0.28 |
| ICHEC-EC-EARTH_RCA4 | + | 0.08 | 1.57 | - | 0.21 | −4.32 |
| IPSL-IPSL-CM5A-MR_WRF331F | + | 0.56 | 0.53 | - | <u>**<0.001**</u> | −24.63 |
| MPI-ESM-LR_CCLM4-8-17 | + | 0.65 | −0.02 | - | <u>**0.03**</u> | −7.87 |
| MPI-ESM-LR_REMO019 | - | 0.87 | −0.21 | - | 0.31 | −3.10 |
| MetEir-ECEARTH_RACMO22E | - | 0.79 | −0.78 | - | <u>**0.01**</u> | −7.60 |
| Project period (2010–2050)—RCP 8.5 | | | | | | |
| CNRM-ARPEGE_CNRM-ALADIN | + | 0.87 | −0.05 | - | 0.61 | −2.04 |
| CNRM-CERFACS-CNRM-CM5_RCA4 | - | 0.37 | −0.28 | - | 0.14 | −5.21 |
| ICHEC-EC-EARTH_RCA4 | + | 0.95 | 0.30 | - | <u>**<0.001**</u> | −23.51 |
| IPSL-IPSL-CM5A-MR_RCA4 | - | <u>**0.02**</u> | −1.66 | - | **0.001** | −11.16 |
| MPI-ESM-LR_CCLM4-8-17 | + | 0.20 | 1.02 | - | <u>**<0.001**</u> | −10.78 |
| MPI-ESM-LR_REMO019 | - | **0.08** | −1.23 | - | <u>**<0.001**</u> | −10.60 |
| MPI-M-MPI-ESM-LR_RCA4 | - | **0.10** | −1.13 | - | **0.06** | −4.40 |
| MetEir-ECEARTH_RACMO22E | + | 0.56 | 0.25 | - | <u>**0.03**</u> | −5.64 |

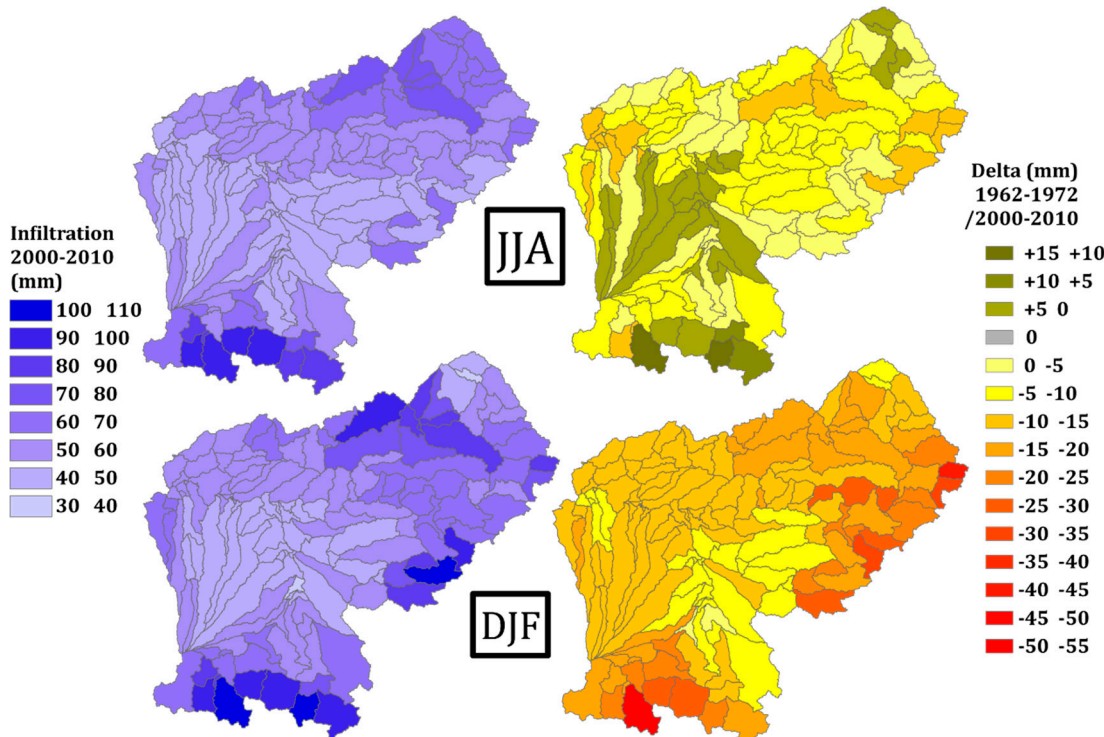

**Figure 9.** Infiltration during the 2000–2010 decade and variations with the 1962–1972 decade.

The subsurface flow is also decreasing over the historical period, with a more significant trend during the summer (Table 10). If looking at Figure 10, the general change of subsurface flows is strongly influenced by some subwatershed in mountainous areas. For instance, a global decreasing trend is visible over winter except for some Pyreneans watershed but yet the Mann–Kendall test does

not return a significant decreasing trend. During summer in those same subwatersheds, subsurface flow is decreasing more than in the rest of the watershed. Projection of subsurface flow seems coherent with observation over the historical period: in winter, no clear trend is appearing when in summer, a clear and significant negative trend is highlighted by Mann–Kendall tests.

**Table 10.** Trend in subsurface flow over winter (DJF) and summer (JJA) seasons at watershed scale, for SAFRAN (historical period) and climatic ensemble (projected period). Bold numbers are significant trend considering α = 0.1 and underlined numbers considering α = 0.05.

| | Subsurface Flow | | | | | |
|---|---|---|---|---|---|---|
| | DJF | | | JJA | | |
| | T (trend) | *P*-Value | Sen's Slope | T (trend) | *P*-Value | Sen's Slope |
| Historical Period (1962–2010) | | | | | | |
| SAFRAN | - | 0.24 | −0.02 | - | **0.019** | −0.02 |
| Project period (2010–2050)—RCP 4.5 | | | | | | |
| CNRM-ARPEGE_CNRM-ALADIN | - | 0.84 | 0.03 | - | **<0.001** | −2.44 |
| CNRM-CERFACS-CNRM-CM5_RCA4 | + | 0.98 | −0.03 | - | **<0.001** | −0.97 |
| CNRM-CM5_CCLM4-8-17 | - | 0.20 | −0.10 | - | **<0.001** | −1.17 |
| ICHEC-EC-EARTH_RCA4 | + | 0.80 | 0.03 | - | **<0.001** | −1.06 |
| IPSL-IPSL-CM5A-MR_WRF331F | - | 0.27 | −0.08 | - | **<0.001** | −0.73 |
| MPI-ESM-LR_CCLM4-8-17 | + | 0.96 | −0.02 | - | **<0.001** | −1.04 |
| MPI-ESM-LR_REMO019 | - | 0.51 | −0.07 | - | **<0.001** | −1.05 |
| MetEir-ECEARTH_RACMO22E | - | 0.53 | −0.11 | - | **<0.001** | −0.92 |
| Project period (2010–2050)—RCP 8.5 | | | | | | |
| CNRM-ARPEGE_CNRM-ALADIN | - | 0.67 | −0.05 | - | **<0.001** | −2.63 |
| CNRM-CERFACS-CNRM-CM5_RCA4 | + | 0.41 | 0.02 | - | **<0.001** | −1.20 |
| ICHEC-EC-EARTH_RCA4 | - | 0.42 | -0.06 | - | **<0.001** | −1.21 |
| IPSL-IPSL-CM5A-MR_RCA4 | + | 0.12 | 0.11 | - | **<0.001** | −0.54 |
| MPI-ESM-LR_CCLM4-8-17 | + | 0.35 | 0.10 | - | **<0.001** | −0.99 |
| MPI-ESM-LR_REMO019 | + | 0.60 | 0.07 | - | **<0.001** | −0.83 |
| MPI-M-MPI-ESM-LR_RCA4 | + | 0.40 | 0.08 | - | **<0.001** | −0.89 |
| MetEir-ECEARTH_RACMO22E | + | 0.99 | −0.04 | - | **<0.001** | −0.99 |

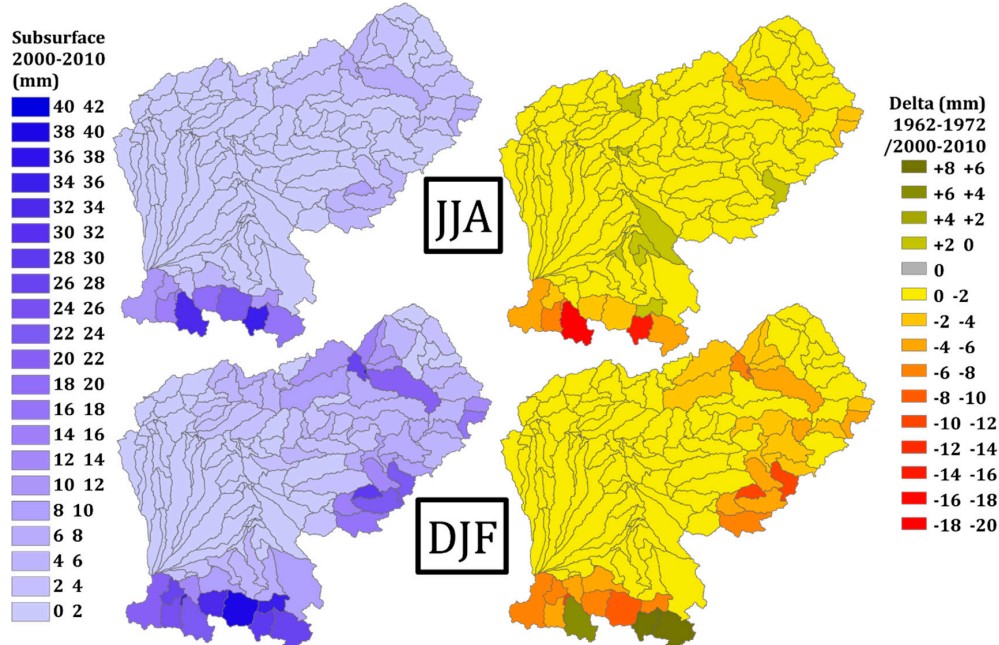

**Figure 10.** Subsurface during the 2000–2010 decade and variations with the 1962–1972 decade.

The two last components of the water cycle examined in this study are green water components. Table 11 and Figure 11 are presenting the evolution of soil water content, impacted by a significant decreasing trend regardless the period, the season or the scenario. This decline of soil water content

appears also homogenous geographically for the last 50 years (Figure 11). This tendency is consistent with the decreasing trend observed for the precipitation and the increasing trend of run off in some subwatershed.

**Table 11.** Trend in soil water content over winter (DJF) and summer (JJA) seasons at watershed scale, for SAFRAN (historical period) and climatic ensemble (projected period). Bold numbers are significant trend considering $\alpha = 0.1$ and underlined numbers considering.

| | Soil Water Content | | | | | |
|---|---|---|---|---|---|---|
| | **DJF** | | | **JJA** | | |
| | **T (trend)** | **P-Value** | **Sen's Slope** | **T (trend)** | **P-Value** | **Sen's Slope** |
| Historical Period (1962–2010) | | | | | | |
| SAFRAN | - | **<0.001** | 0.081 | - | **<0.001** | −9.716 |
| Project period (2010–2050)—RCP 4.5 | | | | | | |
| CNRM-ARPEGE_CNRM-ALADIN | - | **0.05** | −0.10 | - | **<0.001** | −31.14 |
| CNRM-CERFACS-CNRM-CM5_RCA4 | - | **0.02** | −0.22 | - | **<0.001** | −16.86 |
| CNRM-CM5_CCLM4-8-17 | - | **<0.001** | −0.40 | - | **<0.001** | −18.74 |
| ICHEC-EC-EARTH_RCA4 | - | **0.08** | −0.13 | - | **<0.001** | −16.72 |
| IPSL-IPSL-CM5A-MR_WRF331F | - | **0.02** | −0.24 | - | **<0.001** | −14.58 |
| MPI-ESM-LR_CCLM4-8-17 | - | **0.04** | −0.15 | - | **<0.001** | −19.39 |
| MPI-ESM-LR_REMO019 | - | **0.06** | −0.15 | - | **<0.001** | −19.86 |
| MetEir-ECEARTH_RACMO22E | - | **0.002** | −0.47 | - | **<0.001** | −16.31 |
| Project period (2010–2050)—RCP 8.5 | | | | | | |
| CNRM-ARPEGE_CNRM-ALADIN | - | **0.03** | −0.18 | - | **<0.001** | −31.45 |
| CNRM-CERFACS-CNRM-CM5_RCA4 | + | 0.65 | −0.04 | - | **<0.001** | −19.33 |
| ICHEC-EC-EARTH_RCA4 | - | **0.05** | −0.17 | - | **<0.001** | −19.87 |
| IPSL-IPSL-CM5A-MR_RCA4 | - | 0.28 | −0.12 | - | **<0.001** | −14.02 |
| MPI-ESM-LR_CCLM4-8-17 | - | **0.002** | −0.21 | - | **<0.001** | −18.04 |
| MPI-ESM-LR_REMO019 | - | **0.03** | −0.21 | - | **<0.001** | −16.59 |
| MPI-M-MPI-ESM-LR_RCA4 | - | 0.17 | −0.11 | - | **<0.001** | −19.36 |
| MetEir-ECEARTH_RACMO22E | - | **0.07** | −0.36 | - | **<0.001** | −16.42 |

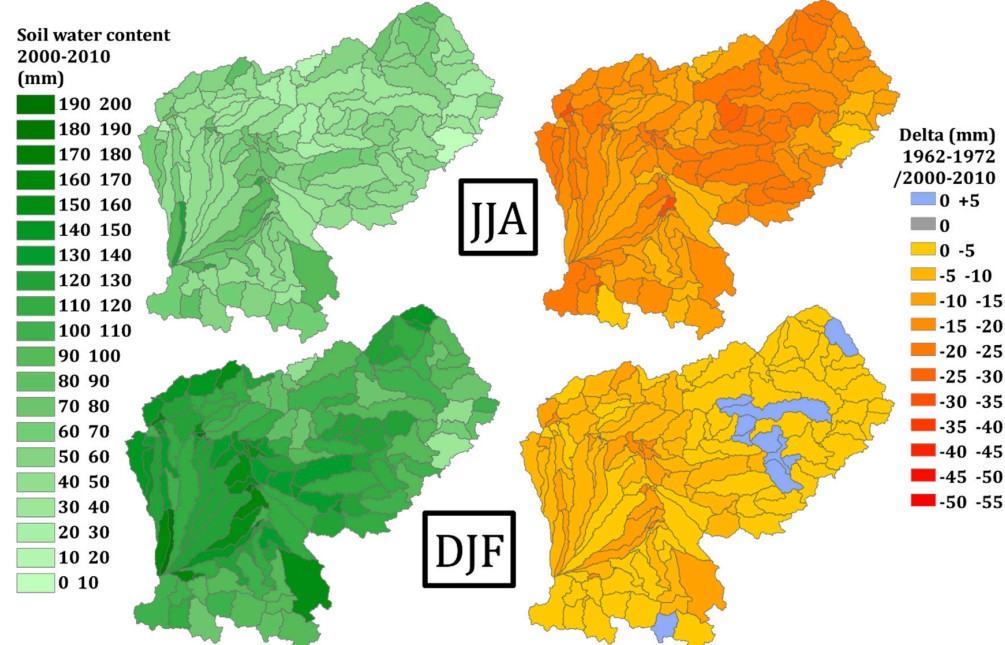

**Figure 11.** Soil water content during the 2000–2010 decade and variations with the 1962–1972 decade.

The soil water content is however highly linked to the evapotranspiration fluxes presented in Table 12 and Figure 12. On both historical and projected periods, trends are similar, showing an increase of evapotranspiration in winter and a decrease in summer. The latter being highly significant, when the winter increase appears significant only for a reduce number of climatic models. Figure 12 shows that this increasing winter trend is covering the overall watershed, when during summer, only

watershed in the plain have a negative delta of evapotranspiration but the trend among subwatersheds located in the mountainous areas remain positive.

**Table 12.** Trend in evapotranspiration over winter (DJF) and summer (JJA) seasons at watershed scale, for SAFRAN (historical period) and climatic ensemble (projected period). Bold numbers are significant trend considering α = 0.1 and underlined numbers considering α = 0.05.

| | Evapotranspiration | | | | | |
|---|---|---|---|---|---|---|
| | **DJF** | | | **JJA** | | |
| | **T (trend)** | ***P*-Value** | **Sen's Slope** | **T (trend)** | ***P*-Value** | **Sen's Slope** |
| **Historical Period (1962–2010)** | | | | | | |
| SAFRAN | + | <u>**<0.001**</u> | 0.081 | - | <u>**<0.001**</u> | −9.72 |
| **Project period (2010–2050)—RCP 4.5** | | | | | | |
| CNRM-ARPEGE_CNRM-ALADIN | - | 0.921 | −0.001 | - | <u>**<0.001**</u> | −9.16 |
| CNRM-CERFACS-CNRM-CM5_RCA4 | + | 0.222 | 0.060 | - | <u>**<0.001**</u> | −12.44 |
| CNRM-CM5_CCLM4-8-17 | + | 0.227 | 0.049 | - | <u>**<0.001**</u> | −12.68 |
| ICHEC-EC-EARTH_RCA4 | + | 0.779 | 0.016 | - | <u>**<0.001**</u> | −16.53 |
| IPSL-IPSL-CM5A-MR_WRF331F | + | **0.093** | 0.051 | - | <u>**<0.001**</u> | −8.22 |
| MPI-ESM-LR_CCLM4-8-17 | + | <u>**0.013**</u> | 0.017 | - | <u>**<0.001**</u> | −15.10 |
| MPI-ESM-LR_REMO019 | + | <u>**0.038**</u> | 0.011 | - | <u>**<0.001**</u> | −15.33 |
| MetEir-ECEARTH_RACMO22E | + | 0.970 | 0.006 | - | <u>**<0.001**</u> | −15.49 |
| **Project period (2010–2050)—RCP 8.5** | | | | | | |
| CNRM-ARPEGE_CNRM-ALADIN | + | 0.648 | 0.017 | - | <u>**<0.001**</u> | −10.65 |
| CNRM-CERFACS-CNRM-CM5_RCA4 | + | 0.682 | 0.004 | - | <u>**<0.001**</u> | −12.76 |
| ICHEC-EC-EARTH_RCA4 | + | <u>**0.007**</u> | 0.054 | - | <u>**<0.001**</u> | −17.51 |
| IPSL-IPSL-CM5A-MR_RCA4 | + | 0.469 | 0.014 | - | <u>**<0.001**</u> | −14.93 |
| MPI-ESM-LR_CCLM4-8-17 | + | <u>**0.022**</u> | 0.017 | - | <u>**<0.001**</u> | −14.60 |
| MPI-ESM-LR_REMO019 | + | <u>**0.035**</u> | 0.011 | - | <u>**<0.001**</u> | −12.54 |
| MPI-M-MPI-ESM-LR_RCA4 | + | 0.654 | 0.010 | - | <u>**<0.001**</u> | −15.40 |
| MetEir-ECEARTH_RACMO22E | - | 0.821 | −0.003 | - | <u>**<0.001**</u> | −15.90 |

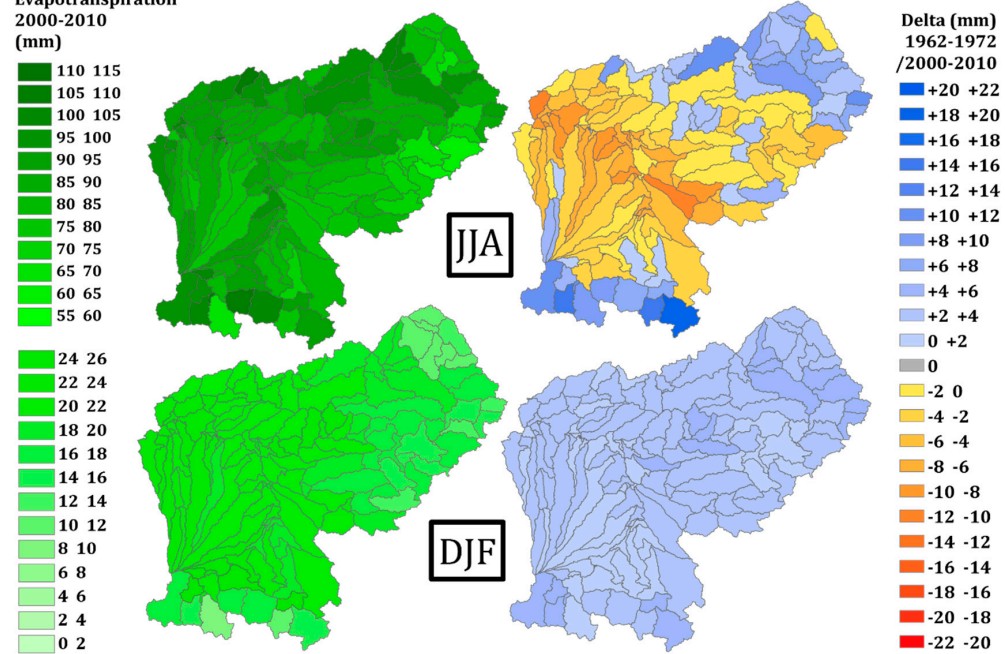

**Figure 12.** Evapotranspiration during the 2000–2010 decade and variations with the 1962–1972 decade.

## 4. Discussion

The performance metrics calculated after the setup of the SWAT model in calibration and validation has confirmed that the present SWAT implementation is functional for long-term hydro-climatic assessment. Only three stations (Foix, Saint-Béat, and Roquefort) located in higher altitudes present slightly lower scores, but the presence of a complex snow dynamics in the upstream

watersheds complicates modelling, as detailed in Grusson, et al. [52], is largely responsible for this discrepancy. Yet SWAT has difficulties in capturing the low flow regime at few locations (lower NSElog values) where some dam operations influence river discharges, namely because of the absence of information on their management strategies. Similar findings were reported by Sauquet, et al. [38] and Hendrickx, et al. [39] who conclude that the impact of existing structures on the discharge quickly diminish beyond the Pyreneans slopes. The models capture well the high discharge and the global volume of water at a monthly time step but face some difficulties to perform during low flow event. The challenge of this study to assess different hydrological component lies in the fact that SWAT is calibrated only with discharge values, since no data were available with a sufficient temporal and spatial resolution for others hydrological component. However, the calibration at a subwatershed scale (each gauging station used to calibrated upstream subwatershed with a dedicated set of parameters) try to handle this problematic by producing a context specific calibration adapted to local hydrological conditions [68,69]. If calibration was not possible to perform for other outputs than discharge, the reliability of the simulated water cycle have been validated using different sources. The balance of simulated hydrological component was positively validated [70] against the physically-based surface hydrological chain of models from Météo-France [71,72]. In mountainous sections, the simulation of snowpack has been compared with punctual data and remote sensing images [52]. In those subwatersheds, runoff is more important, unless the presence of snow during winter increases infiltration and subsurface flows and decreases the direct surface runoff [52]. In addition, tests have been conducted on the same SWAT set-up following the differential split sampling test procedure proposed by Klemeš [73] in order to assess its robustness to climate or land uses changes(presented in Grusson, et al. [74]).

A validation step of the different climate models against observed data has been conducted in this study over the historical period. This step appears primordial to the authors in order to evaluate the possible bias in the representation of regional climate and is yet rarely presented in climate impact assessment. In this study, analysis confirmed that the climatic ensemble was producing a fair representation of the regional climate. The most notable bias is found in very extreme part of the distribution of precipitation (first and last decile). The climate ensemble seems to produce over- and underestimated extremes. This is a known bias of EURO-CORDEX climate models, given the complexity of capturing extreme events [75,76]. On the other hand, this assessment also shown that the SWAT was reacting satisfactorily when inputted with climate data.

Once the modeling chain was deemed suitable to simulate the regional hydrological cycle including if based on climate data projection, several hydrological components have been analyzed over two periods: historical (1962–2010) and future (2010–2050). The choice has been made to present the result focusing on two seasons because of their regional dissimilarity in term of temperature and precipitation [47,77]. The winter snow dynamics in the hydrology of the watershed [36,78] has also been a criteria to consider the output seasonally. The overall analysis of the different hydrological component shows that when the climate ensemble was producing a homogenous and significant trend for the projected period, the same trend were also always visible in the simulations from observed data over the historical period. This consistency points out a steady evolution of the system, and the trends observed during the last 50 years can be expected to persist in the future. The RCP 4.5 scenario and the RCP 8.5 scenario have shown to produce very similar trends for each of the hydrological component. This is consistent with a deeper analysis of both scenario on regional climate projection visible in [79]. Simulated trends appear to be far more influenced by the seasonality than by RCP scenarios.

During winter, mountainous subwatersheds in the Pyrenees are facing a different evolution than the rest of the catchment. They are the only areas where discharge increases from 10 to 20%, in tune with previous studies [10,38,80,81]. It seems to be mostly due to a lower water storage in snowpack as also reported by López-Moreno [82] and Maris, et al. [83]. The effect of snowpack diminution from 20% to 50% and higher liquid precipitation lead to an increase of runoff volume (+10%), a lower infiltration (−20% to −50%) and lower soil water content (−5% to −10%), which is consistent with the study

presented in Grusson, et al. [52]. It should be stressed that the increase of liquid precipitation and the diminution of solid precipitation in altitude during winter is also consistent with previous analyses of the EURO-CORDEX ensemble over France, as reported by Ouzeau, et al. [84] (see also IPCC [85] Annex 1). In those watersheds, the evapotranspiration is increasing during winter, consistently with the rise of temperature and the decreasing of soil water. For the same season, all blue water fluxes in the remaining part of the watershed (hillslope and plain) have been impacted by a general contraction during the historical period. This negative trend does not however appear very significant over the upcoming decades and seems to indicate a stability or a slower decrease. Trends observed from the analysis of the green water components are on the other hand much more significant with a future winter decrease of soil water content and an increase of evapotranspiration, in accordance with the decrease of precipitation and the general increase of temperature.

Precipitation and temperature trends are similar over summer for the historical and projected period, with an increase of temperature and a decrease of precipitation even more significant than for the winter season. The historical period is impacted by a significant diminution of discharge within the catchment (−10 to −30%), with exception of some rare subwatershed in the plain where precipitation increase slightly. This trend is projected to remain the same in the 30 next years and this general contraction of summer discharge volume is consistent with previous study such as Caballero, et al. [81] and Tisseuil, et al. [80]. Others blue water components have also been facing a general decrease over the watershed in the past and projection shown a continuation of this tendency. In the plain, this decline remains limited in volume e.g., runoff with a decrease of about less than 2 mm between 1962–1972 and 2000–2010. Only the discharge seems to be facing a more substantial decrease. The only increase is simulated for few watersheds of the Pyrenean zone, where the model produces an increase of infiltration.

As for winter, a highly significant decline of soil water content impacts the entire watershed during summer, for both historical and projected period. Decreasing of soil water content can reaches 50% in the downstream part of the catchment which is coherent with the literature at the national scale [86,87]. One would thus expect evapotranspiration fluxes to increase, following the increase of temperature, which would stress soil water content. If this is true on mountains and hillsides which seems to be in accordance with the increase of infiltration, in the valley, the evapotranspiration appears to decrease substantially (from −5% to −15%). This decrease indicates periods for which evapotranspiratory demand cannot be met due to a lower soil water availability. The projected evolution is also quite in line with the recent past: the summer soil water content decreases, restricting evapotranspiration, as also reported by Jung, et al. [20] at a global scale. The direct impact of climate change on soil water, evapotranspiration and discharge during summer in southern Europe during the last 50 years is also coherent with Orth, et al. [88].

Projections presented in this study corroborate the evolution reported over the historical period, since most trends are in the same direction. The most notable divergence between recent past and future periods, is the unclear trend for some of surface blue water during winter (runoff and infiltration) when the trend over the historical period is significant. It is also noteworthy that some of those blue water fluxes in few mountainous watersheds are not affected by the same general decreasing trend (e.g., subsurface flow in summer and infiltration in winter). Those hydrological behaviors are not easy to explained, but it must be notice that (i) those subwatersheds were the more complicated to calibrated due to their complex hydrological functioning, and (ii) the analysis presented here report only the total monthly volume and the model setup deployed does not allow to determine if variations originate from an evenly distributed increase or from an increase of extreme event which frequency are also suspected to be modified by climate change [6].

## 5. Conclusions

An integrated approach to assess the impact of climate change on the water cycle at a regional scale has been put together. It aimed understanding variations in the many components of the hydrological

cycle as well as their interconnections, considering the hydrological system as a whole. It offers a guideline to conduct a regional impact assessment, and can be adapted to different hydrological models. The following steps and conclusions are drawn:

The SWAT model was successfully calibrated over a 10-year period (and validated over 40 years) in a step-wise fashion, from the upper part of the watershed to its lower part, using 21 gauging stations in order to encompass local topography and land use and soil diversity. The model was also deemed competent under non-stationary climate, capable of extending the analysis using projected time series issued by a number of GCM/RCM pairs.

An evaluation has been conducted to evaluate the representation of our regional climate offered by the climate ensemble and its influence on the calibration of our hydrological model. Data from climate models were compared with the SAFRAN product over the historical period and the discharge simulated through the SWAT model on the same period compared to observations. A fair representation of the regional climate and the discharge have been shown, allowing us to extend our analysis to future climate.

Many components of the Garonne hydro-system (discharge, snowpack, runoff, infiltration, subsurface flow, evapotranspiration, and soil water content) were identified and analyzed for winter and summer. Changes into the hydrological cycle are assessed by comparing the trend and the geographic spread of variations over the last 50 years with the tendency projected by climate model for the next 30 years. Future trends mostly turned out in the same direction than past ones, suggesting a continuous perturbation of the hydrological system. Future trends are found more divergent between seasons than between the two projected scenarios (RCP 4.5 RCP 8.5). If we look the watershed globally and for both seasons, blue water fluxes are expected to decrease over the watershed, with a more significant trend for summer than winter. Discharge is the blue water fluxes which have been and are expected to be further impacted by the strongest decreasing trend. Similarly, the soil water is facing a significant decreasing trend through the entire reported period, while the evapotranspiratory demand is found increasing. Two important seasonal dissimilarities within the hydrological behaviors of the watershed have been highlighted. In winter, discharge increase in higher grounds because of lower snowpack associated to an increase of surface runoff for some subwatershed. During the same period, soil water content is decreasing, but can still support the evapotranspiratory demand, leading to an actual increase of evapotranspiration. In summer, however, the decrease of blue water fluxes is more comprehensive, and the soil water content level in the plain does not allow the evapotranspiratory demand to be met, and the evapotranspiration is actually decreasing. Only a few subwatersheds in the mountainous area can respond to this demand generated by an increase of temperature, and support an increasing evapotranspiration.

Further analysis at shorter temporal scales could offer more information of the repartition of those fluxes over time, regarding the occurrence of extreme precipitation events.

**Author Contributions:** Y.G. conceived, designed and performed the experiments and wrote the paper. F.A. and J.M.S.P. supervised the research activities. S.S. also provided help throughout the work. All coauthors have collaborated on the redaction of the manuscript.

**Funding:** This research was funded by the Natural Sciences and Engineering Research Council of Canada and the Institut Hydro-Québec en environnement, développement et société. This research was carried out as a part of ADAPT'EAU (ANR-11-CEPL-008), a project supported by the French National Research Agency (ANR) within the framework of the Global Environmental Changes and Societies (GEC&S) program. This work was also part of the REGARD project (Modélisation des ressources en eau sur le bassin de la Garonne: interaction entre les composantes naturelles et anthropiques et apport de la télédétection)—RTRA Sciences et Technologies pour l'Aéronautique et l'Espace—2014–2017.

**Acknowledgments:** We sincerely thank Météo-France for providing meteorological data and AEAG (Agence de l'eau Adour-Garonne) for providing hydrological discharge data. Authors also thanks the help offered by DRIAS services (Météo-France/IPSL/CERFACS) to access climate data (http://www.drias-climat.meteo.fr).

**Conflicts of Interest:** The authors declare no conflict of interest.

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
