# Peer review of "Coevolution of Hydrological Cycle Components under Climate Change: The Case of the Garonne River in France"

_water, doi:10.3390/w10121870_

Reviewer 1 Report

Authors did a tremendous job in this manuscript, and definitely has high merits to be published. But before being ready for the publication, I recommend minor revision for following points:

English grammar for the whole manuscript need to be thoroughly checked. You can find some of the comments in the attached file. 

Although you have given good background, the point of uniqueness is somewhat missing in your introduction section. Once added one paragraph it will be making this paper very interesting for the readers.

In result and discussion section, the reason for poor performance (both calibration and validation) of the model at some of the stations is not well described. Please write it more clearly.

Author Response

·         “English grammar for the whole manuscript need to be thoroughly checked. You can find some of the comments in the attached file.”
An overall checked has beenconducted over the manuscript by several of the co-authors.

·         “Although you have given good background, the point of uniqueness is somewhat missing in your introduction section. Once added one paragraph it will be making this paper very interesting for the readers.”  
We added a sentence following your advice: “In short, many advocate exploring the coevolution of all major components of the water cycle (and their different water paths) but such endeavor is still rarely conducted.”

·         “In result and discussion section, the reason for poor performance (both calibration and validation) of the model at some of the stations is not well described. Please write it more clearly."

We added the following sentences to the discussion: “Only three stations (Foix, Saint-Béat and Roquefort) located in higher altitudes present slightly lower scores, but the presence of a complex snow dynamics in the upstream watersheds complicates modelling, as detailed in Grusson et al. [52] is largely responsible for this discrepancy. Yet SWAT has difficulties in capturing the low flow regime at few locations (lower NSElog values) where some dam operations influence river discharges, namely because of the absence of information on their management strategies. Similar findings were reported by Sauquet, et al. [38] and Hendrickx, et al. [39] who conclude that the impact of existing structures on the discharge quickly diminish beyond the Pyreneans slopes.”  

            ·         Comment Line 82: These values are monthly inter-annual means. The text has been modified accordingly for more clarity. 

            ·         Comment on Figure 1 (line 85): In Figure 1, the watersheds are delimited by the thick black lines. We think that the confusion comes from 

            illustrating the river network using a dashed blue line, we change it for a thick blue line for the river network. Figure 2 has been modified accordingly.

            ·         Comment Line 121: Has been modified.

            ·         Comment Line 226: Explanation were already presented in the discussion part. Those explanation have been developed (see Line 390-394)

            ·         Comment on Table 5: Has been modified.

            ·         Comment Line 316: Has been modified.

            ·         Comment Line 322: Has been modified.

Reviewer 2 Report

Reviewer’s Report

Journal: Water

Manuscript ID: water-406829

Title:  Coevolution of hydrological cycle components under climate change: The case of the Garonne River in France

Authors: Youen Grusson , François Anctil , Sabine Sauvage , José Miguel Sánchez Pérez

General comments

This study applied a SWAT model to assess the evolution of the many hydrological variables of the Garonne basin, France. I think this article is well organized to read and the contribution and methods of this study is very suitable to assess the effect of the climate change. Therefore, I think this article is acceptable to be published in Water journal.

Author Response

Reviewer 2

·         “This study applied a SWAT model to assess the evolution of the many hydrological variables of the Garonne basin, France. I think this article is well organized to read and the contribution and methods of this study is very suitable to assess the effect of the climate change. Therefore, I think this article is acceptable to be published in Water journal.”

No response to provide.

Reviewer 3 Report

I found the manuscript interesting. I found the data period sufficient. The SWAT model calibrated and validated in an appropriate way and the manuscript is overall in a high scientific status. 

Author Response

·         “I found the manuscript interesting. I found the data period sufficient. The SWAT model calibrated and validated in an appropriate way and the manuscript is overall in a high scientific status.”

No response to provide.